

**Response of soil nutrients and erodibility to slope aspect in the northern**
**agro-pastoral ecotone, China**
Yuxin Wu[a,b,c], Guodong Jia[a,b,c,*], Xinxiao Yu[a,b,c*], Honghong Rao[d], Xiuwen Peng[e],
Yusong Wang[a,b,c]
[a]Key Laboratory of State Forestry and Grassland Administration on Soil and Water
Conservation, Beijing Forestry University, Beijing 100083, PR China
[b]The Metropolitan Area Forest Ecosystem Research Station, School of Soil and Water
Conservation, Beijing Forestry University, Beijing 100083, PR China
[c]The Metropolitan Area Field Scientific Observation Research Station, School of Soil
and Water Conservation, Beijing Forestry University, Beijing 100083, PR China
[d]Department of Education, Tianjin Normal University, Tianjin 300387, PR China
[e]Shanghai Investigation, Design & Research Institute Co., Ltd, Shanghai 200126, PR
China
* Corresponding author. Address: No.35 Tsinghua East Road, Haidian District,
Beijing Forestry University, 100083 Beijing, China.
Email address: jiaguodong1111@163.com(G.Jia). yuxinxiao11111@163.com(X.Yu).
**Abstract**
Soil erosion, considered a major environmental and social problem, leads to the
loss of soil nutrients and the degradation of soil structure, impacting plant growth.
However, data on the effects of land use changes caused by vegetation restoration on
soil nutrients and erodibility at different slope aspects is limited. This study was
conducted to detect the response of soil nutrients and erodibility of different slope
aspects in a typical watershed of the northern agro-pastoral ecotone in China. The
following indexes were used to determine the improvement of soil nutrients and
erodibility through a weighted summation method: comprehensive soil nutrient index
and comprehensive soil erodibility index. The results showed that the vegetation types
with the highest comprehensive soil quality index (CSQI) on the western, northern,
southern, and eastern slopes were *Pinus sylvestris* and *Astragalus melilotoides* (1.45),
*Caragana korshinskii* and *Capillipedium parviflorum* (2.35), *Astragalus melilotoides*
(4.78), and *Caragana korshinskii* and *Lespedeza bicolor* (5.00), respectively. Slope
aspect had a significant effect on understory vegetation characteristics, soil nutrients,
and soil erodibility. Understory vegetation and soil characteristics could explain
50.86–74.56% of the total variance in soil nutrients and erodibility of slope aspect.
Mean weight diameter, total phosphorus, saturated hydraulic conductivity, and soil



disintegration rate were the main factors affecting CSQI on different slope aspects.
Our study suggested the combinations of species, such as *C. korshinskii* and *L. bicolor*,
were the best species to include on any slope aspect in regards to improving soil
nutrients and soil erodibility.
**Keywords:** Slope aspect; Soil nutrients; Soil erodibility; Soil erosion; Vegetation
restoration; Land use
**1. Introduction**

Soil erosion, considered a major environmental and social problem, leads to the

loss of soil nutrients and the degradation of soil structure, influencing the functional
capacity of soils on a global scale (Singh and Panda, 2017; Wen et al., 2021).
Vegetation restoration is an important method of ecological restoration that aims to
control soil erosion and prevent soil degradation (Schmiedel et al., 2017; Zhang et al.,
2021). Vegetation restoration can improve the soil structure and nutrients, which in
turn promotes the restoration of soil quality and function (Guo et al., 2021; Li et al.,
2017). Changes in land use due to vegetation restoration play an important role in
improving the environment and ecosystem function, as well as improving soil quality
and soil nutrient cycling (Akiyama and Kawamura, 2007; Singh and Gupta, 2018).

Previous studies have shown that the plants selected for vegetation restoration

projects drive land use change and alter soil properties, thus affecting soil erodibility
(Wang et al., 2019b, a; Zhang et al., 2019). Many studies have also elucidated the
influences of land use change on soil nutrients and have confirmed that revegetation is
an effective way to enhance soil nutrients (Huang et al., 2020; Li et al., 2020; Yang et
al., 2021; Zhu et al., 2020). However, it is not clear which plants selected for
restoration are the most effective in enhancing soil nutrients and reducing soil
erodibility. Most studies have only focused on one aspect; thus, they lack
comprehensive consideration and evaluation of the impact of land use changes caused
by vegetation restoration on soil nutrients and erodibility. The lack of a
comprehensive understanding prevents us from gaining the best ecological benefits
from vegetation restoration. Therefore, studies must be conducted on the response of
soil nutrients and erodibility to different vegetation restoration types.

Soil erodibility is the sensitivity of the soil surface to erosion processes (Batista

et al., 2023; Bryan et al., 1989). It is a necessary parameter for establishing soil loss
equations and erosion models. There is currently no soil erosion model that can
accurately predict soil erosion, although there are many related models (de Vente et al.,





2013, 2008). At present, the soil erodibility K-factor, as defined in the general soil
loss equation (USLE), is the most widely used measure (Wischmeier and Smith,
1978). In addition to K, other soil indexes have been adopted, including saturated
hydraulic conductivity (SHC), soil disintegration rate (SDR), mean weight diameter
(MWD), soil structural stability index (SSSI), clay ratio (CR), and soil organic carbon
cementing agent index (SCAI), to quantify soil erodibility (Dong et al., 2022a; Guo et
al., 2021; Wang et al., 2018; Zhang et al., 2019). Soil organic carbon, nitrogen, and
phosphorus as well as their stoichiometry is also essential for assessing soil quality as
well as ecosystem productivity and functionality (Borchard et al., 2017; Li et al., 2020;
Masciandaro and Ceccanti, 1999; Schloter et al., 2003). A single index cannot fully
reflect all soil properties; therefore, it is necessary to develop a comprehensive soil
index using several related indicators.
In addition to soil properties, topographic factors also significantly affect soil
nutrients and erodibility (Bangroo et al., 2017; Nabiollahi et al., 2018; Qin et al., 2016;
Zhang et al., 2018). Slope aspect can affect the growth of plants due to a combination
of factors, such as light, temperature, wind speed, and precipitation, which can cause
significant changes in the ecological relationship between plants and the environment
(Li et al., 2018; Tamene et al., 2020; Zhang et al., 2020). This is especially true for
harsh climates such as cold, dry alpine regions in the north, in which plants are more
sensitive to environmental changes. However, the optimal vegetation restoration type
has primarily been studied by slope gradient and slope position (Dong et al., 2022a;
Guo et al., 2021; Wen et al., 2021). There is a lack of systematic evaluation of the
effects of land use changes caused by vegetation restoration on soil nutrients and
erodibility on different slope aspects. Therefore, the classification of slope aspect
needs to be further refined to elucidate the response of different slope aspects to
changes in soil nutrients and erodibility caused by revegetation.
The ecologically fragile northern agro-pastoral zone in China is located in an
erosion zone affected by both wind and water; soil erosion in this zone is considered
very serious (Guo et al., 2019). Recently, the Chinese government has planned and
carried out a series of ecological restoration projects in this region, including the
Beijing-Tianjin Wind and Sand Source Control Project, the Beijing-Hebei Water
Protection Forest Project, and the Sebei Forest Plantation Afforestation Project. These
ecological restoration projects have effectively reduced land erosion and
desertification, and have significantly delayed the onslaught of wind and sand (Wang



et al., 2021b; Zeng et al., 2014; Zhang et al., 2017). However, the method used for
afforestation, which mainly consists of plantations, is affected by differences in water,
heat, wind, and sand in the different habitats, making it difficult to achieve vegetation
restoration in some ecologically fragile areas, and the selection of suitable tree species
is still equivocal.

Based on the abovementioned scientific gaps, we hypothesize the following: 1)

vegetation restoration can significantly alter soil structure and properties to influence
soil nutrients and erodibility during the process of vegetation restoration; 2) both
slope aspect and land use types can significantly affect soil nutrients and erodibility; 3)
the western slope may have the lowest comprehensive soil quality index compared to
other slope aspects. Therefore, we selected four slope aspects (west, north, south, and
east) that have four different land use types (degraded land, grasslands, shrublands
and woodlands) in a typical watershed of the northern agro-pastoral ecotone with
three specific purposes: 1) to determine the impact of different vegetation types on
different slope aspects on soil nutrient improvement and soil erodibility enhancement;
2) to determine the key influencing factors affecting soil nutrients and erodibility of
the four slope aspects; and 3) to provide optimal revegetation models for improving
soil nutrients and reducing soil erodibility on different slope aspects.
**2.  Materials and Methods**
**2.1. Study area**

This study was conducted in the Yangcaogou Watershed (41°4′~41°8′ N,

114°58′~115°2′ E; Fig.1), Chongli District, Zhangjiakou City, Heibei Province, China.
The watershed is located in a typical ecological transition zone of the agro-pastoral
ecotone in northern China (Wu et al., 2023). The study site spans an area of 10.6 km$^2$
with an altitude ranging from 1084 to 1575 m. It belongs to a typical temperate
continental monsoon semi-arid climate with an annual average temperature of 3.5 ℃.
The average annual rainfall is 401.6 mm. The rainy season occurs from June to
September (Chang et al., 2021; Guo et al., 2019). The main soil type is classified as
chestnut soil in both the Chinese Soil Taxonomy and the World Reference Base for
Soil Resources (Schad, 2017). Most of the study area consists of Proterozoic soil rock
formations. Owing to irrational human reclamation and grazing, there is very serious
soil and gully erosion. Over the past decade, due to the implementation of the
Beijing–Tianjin Sandstorm Source Control Project, soil erosion and desertification
has been effectively mitigated (Wang et al., 2020b). However, native plant





populations have been diminished and instead the area is planted with trees, shrubs,
and herbs.

**2.2. Selection of sites and determination of slope aspect**

The study was conducted during the 2021 growing season. A comprehensive
field survey was conducted on the dominant plant species and soil properties of each
of the following land use types: degraded land, grasslands, shrublands, and woodlands
in the Yangcaogou watershed. Grasslands, shrublands, and woodlands were restored
from degraded land over the past 12 years. The degraded land was previously
degraded cropland. All land use types were vegetated and restored in the form of
engineering measures such as fish scale pits (Wang et al., 2014b) and parallel ditches
(Barua and Alam, 2013).
In addition to the degraded land, the other three land use types were all sampled
along complete slope aspects at the E, W, N, and S slopes. It includes 28 sample sites
(20 m × 20 m) of an degraded land, two grasslands, two shrublands and two
woodlands on each slope aspect. Three sampling quadrats (1 m × 1 m) were set up
in each sample site to investigate and record the species, height, richness, coverage,
aboveground biomass, belowground biomass, and litter biomass of herbs. Height was
measured as the average height of herbs in the sample. Biomass coverage was
determined following the visual method (Proulx and Mazumder, 1998). Richness was
calculated by measuring the number of individuals of each herb in the quadrat and
calculating the percentage of its occurrence (Dou et al., 2023). Belowground biomass
and soil samples were collected with a 9 cm diameter soil drill. The measured land
use types, major plant species, and understory vegetation characteristics at each
selected field site are listed in Table S1.
Following the methods described by (Yimer et al., 2006), study sites were
selected that included the four land use types on each of the four slope aspects: east,
west, north, and south. Eastern, western, northern, and southern slopes are also known
as semi-sunny, semi-shady, shady, and sunny slopes(Che et al., 2022; Chen et al.,
2021b). In this region, it is difficult to find degraded land because the vast majority of
the degraded land had been converted to artificial forest and grass vegetation.
Therefore, four unrestored degraded land were selected as representatives from the
western slope. The slope gradients and positions were similar for all selected sample
sites (Fig. 1).



### 2.3. Soil sampling and analysis

Three quadrants were selected at each site to investigate vegetation and collect soil samples. For each sampling point, a steel cutting ring (100 cm$^3$) was used to obtain 75 soil samples (25 sites × three sampling points). The saturated hydraulic conductivity of the soil were evaluated using the constant head permeability test (Chandler and Chappell, 2008). The mean weight diameter was measured by screens with different pore sizes (0.25, 0.50, 1.00, 2.50 and 5.00 mm) (Campo et al., 2008). After air-drying via dry screening, 50 g of the soil samples were placed on the sieve of a soil aggregate analyzer (TTF-100 model, China), then completely immersed in water, and shaken up and down 30 times for 1 minute (Wang et al., 2014a). After shaking, samples were removed from the settling cylinder, and the remaining aggregates on each sieve were put into an aluminum box for drying. Finally, the samples were weighed and the dried aggregates were recorded.

Soil characteristics of different vegetation types at different slope aspects are listed in Table S2. Topsoil samples were collected from 0–10 cm using a cutting ring. Samples were brought back to the lab to oven-dried at 105℃ for 24 hours. Then, the soil bulk density (SBD) (Lardy et al., 2022; Moreira et al., 2020) and soil capillary porosity (SCP) (Singh and Pollard, 1958) were measured. In addition, 225 mixed soil samples (25 sites × three quadrats/site × three samples/quadrat) were collected as soil samples. Among them, the particle size distribution of clay content (Cl), silt content (Si), sand content (Sa) was determined by a Microtrac S3500 laser particle sizer (Malvern 3000, UK). Total nitrogen (TN) and total phosphorus (TP) were determined by the dichromate oxidation (Bremner, 1996) and HClO$_4$-H$_2$SO$_4$ methods (Kisand, 2005), respectively. Soil pH (Cornfield, 1954) was determined using a pH meter at a 2.5 soil:1 water ratio.

### 2.4 Calculation of soil indexes

Saturated hydraulic conductivity of the soil (K$_S$) (Campo et al., 2008), mean weight diameter (MWD) (Ortas and Lal, 2012), soil disintegration rate (SDR) (Guo et al., 2021), soil structure stability index (SSSI) (Nichols and Toro, 2011), soil organic carbon cementing agent index (SCAI) (Dong et al., 2022a) and K factor (Jiang et al., 2020; Li et al., 2012) were used to express the soil erodibility. These indexes were calculated using equations (1) - (5):

$$K_S = \frac{QL}{Aht} \tag{1}$$



where Q is the outflow volume (ml), A is the soil column section (mm$^2$), t is the time
(min), h is the head difference (mm), and L is the height of the soil column (mm).
$$MWD = \sum_{i=1}^{n} (w_i/m_t)d_i \qquad (2)$$
Where $w_i$ is the mass of the i-th level of aggregates or other soil material (g), $m_t$ is
the sample mass, and $d_i$ is the mean diameter of the i-th level of aggregates or other
soil material (mm).
$$SDR = \frac{M_1 - M_2}{t_2 - t_1} \times 100\% \qquad (3)$$
Where $M_1$ and $M_2$ are the weight of the soil before ($t_1$) and after ($t_2$) disintegration,
respectively.
$$SSSI = 100\% \times \frac{SOMC}{Cl+Si} \qquad (4)$$
$$K = \left\{0.2 + 0.3 \exp\left[-0.0256 Sa\left(1 - \frac{Si}{100}\right)\right]\right\} \left(\frac{Si}{Cl+Si}\right)^{0.3} \times \left(1 - \frac{0.25C}{C + \exp(3.72 - 2.95C)}\right)\left(1.0 - \frac{0.7SN1}{SN1 + \exp(-5.51 + 22.9SN1)}\right)$$
$$(5)$$
Where SOMC is the content of soil organic matter (Kar et al., 2023), C = 0.583 ×
SOMC; Cl and Si represent the clay and silt content (%), respectively; SN1 =
1-Sa/100; K represents the soil loss rate per unit area under rainfall erosivity
conditions for a specified soil on a standard plot (Jiang et al., 2020; Renard et al.,
1997). A previous study indicates the rationality and validity of estimating K in the
Zhangjiakou region using this model (Wang et al., 2020a).

In order to further evaluate soil nutrients and erodibility, comprehensive soil

nutrient and erodibility index were calculated using equations 6 and 7, respectively:
$$CSNI = \sum_{i}^{n} K_{ni} \cdot C_{ni} \qquad (6)$$
$$CSEI = \sum_{i}^{n} K_{ei} \cdot C_{ei} \qquad (7)$$
Where $K_{ni}$ and $C_{ni}$ are the weight and score of soil nutrient index respectively, $K_{ei}$ and
$C_{ei}$ are the weight and score of soil erodibility index respectively, and n is the number
of indexes.

The weight of each soil nutrient index and soil erodibility index was determined

using a principal component analysis (PCA) (Pandey et al., 2021; Wang et al., 2018).
The scores of SHC, MWD, SSSI, SOC, TN, and TP scores were calculated using a
"reverse S" function, which was calculated using equations 8.



$$f(x) = \begin{cases} 1 & , x \geq b \\ \frac{x-a}{b-a} & , a < x < b \\ 0 & , x \leq a \end{cases} \qquad (8)$$
The SDR and K factor scores were calculated by "S" function, as shown in
equations 9.
$$f(x) = \begin{cases} 1 & , x \leq b \\ \frac{x-a}{b-a} & , a > x > b \\ 0 & , x \geq a \end{cases} \qquad (9)$$
Comprehensive soil quality index (CSQI) is used to express soil quality, which
takes into account both soil nutrients and erodibility (De Laurentiis et al., 2019; Dong
et al., 2022b). The CSQI was calculated as follows (Eq. 10):
$$CSQI = \frac{CSNI}{CSEI} \qquad (10)$$
where CSQI ($>$ 0), CSNI (0-1) and CSEI (0-1) are the comprehensive soil quality,
nutrient, and erodibility indexes, respectively.
**2.5. Statistical analysis**
Excel 2016 and SPSS Ver. 20 software were used for data processing and
statistical analysis, and ArcGIS 10.4.1 and Origin 2021 were used for graphing. A
one-way analysis of variance (ANOVA) was used to compare soil nutrient and
erodibility indexes of different slope aspects and different land use types. The effects
of land use types, slope aspects and their interaction on soil nutrients and erodibility
indexes were tested using a two-way ANOVA. Pearson's correlation coefficient was
used to determine the correlation between soil nutrient, erodibility, and quality
indexes and their influencing factors. The contributions of understory vegetation and
soil characteristics to total variance in soil nutrients and erodibility indicators were
determined using a redundancy analysis (RDA) (Capblancq et al., 2018; Peres-Neto et
al., 2006). A random forest algorithm based on R software was used to analyze the
importance of impact factors from different slope aspects (Schonlau and Zou, 2020;
Vincenzi et al., 2011). The importance index was determined as the average accuracy
reduction. When the importance index is higher, it means that the corresponding
factor holds more weight (Chen et al., 2021a; Hao et al., 2015).
**3. Results**
**3.1. Changes in the characteristics of understory vegetation on different slope**



**aspects**

Slope aspect significantly influenced some of the characteristics of understory vegetation such as aboveground biomass (AGB) and belowground biomass (BGB). All measured characteristics of understory vegetation on the western slope were lower than that of other three slope aspects. AGB and BGB was significantly lower for the western slope than the eastern slope (Fig. 2). AGB and BGB on the eastern slope were significantly higher than those on the western slope by 63.40% and 78.40%, respectively (Fig. 2d, e). The measured plant characteristics from the eastern and western slopes were not significantly different from those on the northern and southern slopes. There were significant differences among the four land use types for all characteristics measured for the western slope (Table S1). BH, R, and AGB of understory vegetation were significantly higher for the woodland than for the other three land use types (Fig. 2). Overall, shrubland had the highest litter biomass on each slope aspect, while degraded land on the western slope had the lowest.

**3.2. Changes in soil nutrients on different slope aspects**

Slope aspect significantly affected soil nutrients. Soil organic carbon (SOC), total nitrogen (TN), and total phosphorus (TP) were significantly lower in soil collected from the western slope than the eastern slope (Fig. 2). SOC of the eastern slope was 0.96–1.38 times greater than that of other slopes, respectively (Fig. 2g). TN was highest on the eastern slope and was 0.39 g kg$^{-1}$ and 0.28 g kg$^{-1}$ greater than that on the western and northern slopes, respectively (Fig. 2h). Similarly, the TP of the eastern slope was significantly greater than that of the southern and eastern slopes by 59.60% and 17.37%, respectively (Fig. 2i). When all slope aspects were considered, comprehensive soil nutrient index (CSNI) was significantly lower on the western slope than on the other three slope aspects. The highest CSNI was found for both southern slope (0.81) and eastern slope (0.86) (Fig. 3). For a given slope aspect, land use types also significantly influenced soil nutrients (Fig. S1). For exemple, on the western slope, the SOC of forested land was significantly higher than other restored land uses by 11.81–150.84% depending on the comparison. SOC, TN, and TP of degraded land were significantly lower than that of other land use types. CSNI was influenced by land use type, slope aspect, and their interactions (Table 1). Compared to degraded land, CSNI was significantly higher for all three land uses, with the greatest increase in CSNI for shrubland (0.75), followed by woodland and grassland (Fig. 4).



### 3.3. Changes in soil erodibility under vegetation restoration

The effect of slope aspect on soil erodibility indicators was significant (Table 1 and 2). Among the four slope aspects, SHC of the soil collected from the eastern slope was the greatest, and was significantly greater than that of the western and northern slopes by 311.16% and 187.10%, respectively. MWD was highest on the eastern slope (3.65 mm), followed by the southern and northern slopes. MWD among the four slopes was significantly different. SSSI of the western slope was the lowest (0.41 g kg$^{-1}$), and it was significantly lower than the other three slope aspects. In contrast, the highest SCAI was found on the western slope, and it was significantly higher than the other slope aspects by 46.10%–59.70%, respectively. When all slope aspects were considered, the southern (0.26) and eastern (0.20) slopes had the highest comprehensive soil erodibility index (CSEI) reduction capacity (Fig. 3). For any given slope aspect, land use types also greatly influenced soil erodibility indicators (Table 2). On the western slope, MWD was significantly increased by 0.67 mm–1.59 mm. On the northern slope, the SHC of woodland was significantly higher than that of shrubland (by 117.67%) and grassland (by 94.24%), respectively. On the southern slope, the K in the grassland land use type was significantly lower than that in woodland and shrubland. On the eastern slope, soil disintegration rates of the three restored land uses were significantly different, with the highest SDR in the woodlands. CSEI was influenced by land use type, slope aspect, and their interactions (Table 1). The CSEI of all three restored land uses was significantly lower by (63.01–64.70%) compared to the degraded land (Fig. 4).

### 3.4. Changes in comprehensive soil quality index under vegetation restoration

When all slope aspects are considered, there were significant differences in comprehensive soil quality index (CSQI), with the eastern slope (2.46) having the greatest capacity to increase CSQI (Fig. 3). Compared to degraded land, the CSQI of grassland, shrubland and woodland increased significantly by 2.51, 2.65, and 2.44, respectively (Fig. 4). CSQI was influenced by land use type, slope aspect, and their interactions (Table 1).

The differences in CSQI of different vegetation types were compared to determine the optimal vegetation restoration type for different slope aspects. On the western slope, the WGCP grassland (*Capillipedium parviflorum*) and WWPS woodland (*Pinus sylvestris* and *Astragalus melilotoides*) had relatively high CSQIs. They were significantly higher than that of other vegetation types (Fig. 5a). Therefore,





these two plant communities may be selected for restoration practices on the western
slope. On the northern slope, the CSQI of the shrubland (NSCK) was significantly
higher and second highest in grassland (NGBI). The combination of *Caragana*
*korshinskii* and *Capillipedium parviflorum* (NSCK) could also be selected as taxa for
restoration vegetation (Fig. 5b). On the southern slope, the CSQI of grassland (SGAM)
was significantly higher than that of other vegetation types (Fig. 5c). The SGAM was
dominated by the herb *Astragalus melilotoides*, which had the highest CSQI. *A.*
*melilotoides* could be selected for improving soil quality on the southern slope. On the
eastern slope, the CSQI of the shrubland (ESCK) was relatively higher than that of
other sites (Fig. 5d). The ESCK was dominated by *Caragana korshinskii* and
*Lespedeza bicolor*,which had the highest CSQI. Therefore, these species should be
selected for improving soil quality on the eastern slope.

**3.5. Key factors and their contributions on different slope aspects**

The RDA followed by Monte Carlo permutation tests revealed that the variations
in the nine measured soil quality indicators were significantly influenced by
understory vegetation and soil characteristics on the four slope aspects ($P < 0.01$, Fig.
6). On the western slope, 62.7% of the total variance can be explained by understory
vegetation and soil characteristics (Fig. 6a), with understory vegetation and soil
characteristics explaining 43.11% and 19.59% of the total variance, respectively. For
the northern slope, the understory vegetation and soil characteristics contributed
50.86% of the total variance of soil quality (Fig. 6b), of which understory vegetation
and soil characteristics accounted 33.28% and 17.58% of the total variance,
respectively. On the southern slope, the total variance in soil quality of 54.23% could
be explained by understory vegetation and soil characteristics, of which the
combination of soil and roots contributed 44.56% and 9.67% of total variance,
respectively (Fig. 6c). However, on the eastern slope, the understory vegetation and
soil characteristics contributed 74.56% of the total variance of soil quality (Fig. 6d),
of which understory vegetation and soil characteristics accounted for 56.81% and
17.59% of the total variance, respectively.
The random forest analysis highlighted the importance of 21 modeling factors to
determine the restoration characteristics of understory vegetation and the physical and
chemical characteristics of topsoil on different slope aspects. MWD, TP, saturated
hydraulic conductivity (SHC), and soil disintegration rate (SDR) were the main
factors influencing understory vegetation and soil properties on different slope aspects.



The mean accuracy reduction was calculated using the random forest method. Using
this calculation, we obtained an MWD of 13.40, TP of 13.30, SHC of 12.60, and SDR
of 8.20 (Fig. S2).
**4. Discussion**
**4.1. Effects of slope aspect on understory vegetation characteristics**
Slope aspect, one of the most important topographic factors, may impacts
vegetation characteristics due to differences in sunlight, moisture, temperature, and
soil (Fig. 2). Soil is the material basis for plant growth, and there is an important
relationship between plant growth, development, and distribution and the soil
characteristics of different slope aspects (Gao, 2017; Zhou et al., 2020). There is a
synergistic evolutionary and adaptive relationship between plant growth and survival
in the environment. Moreover, plants grow differently on different slope aspects,
showing plastic responses depending on their habitat (Che et al., 2022; Sharma et al.,
2010).

Our results showed that most of the characteristics of understory vegetation had
no significant differences based on the different slope aspects. This may be due to the
fact that the understory plants were shaded by the taller trees and shrubs (Niinemets,
2010). Aboveground biomass was greater on the eastern and southern slopes than on
the northern and western slopes. Vegetation density was lowest on the western slope.
These findings indicated that aboveground biomass is closely related to sunshine
hours. Sunshine hours affect the balance of heat and water (Chen et al., 2021b; Shi et
al., 2021). This contributed to the low aboveground biomass of the western slope.
Similarly, belowground biomass declined from the eastern, southern, northern, and
western slopes. This may be due to the difference in the aboveground biomass of the
four slope aspects. Aboveground biomass impacts belowground biomass (Sun et al.,
2022), and the belowground biomass was significantly lower on the western slope
than on the eastern slope.
In view of the influence of slope aspect on the establishment of restored
vegetation in the study area, the number of seedlings on the western and northern
(shaded) slopes should be increased at the early stage of vegetation restoration in the
northern agro-pastoral ecotone. In addition, timely replanting and follow-up
application of nitrogen fertilizer during the restoration process will help to reduce the
differences in vegetation growth caused by the inherent differences among the slope
aspects.



**4.2. Effects of slope aspect on soil nutrients**

Soil nutrients play an important role in the maintenance and improvement of soil quality. Soil nutrients are an important reflection of the ecological effects of vegetation restoration (Salekin et al., 2021; Wang et al., 2012; Yüksek and Yüksek, 2021). Our results show that the conditions related to slope aspect have significant effects on single soil nutrient indicators and the comprehensive soil nutrient index (Figs. 2, 5). In the same area, soil nutrients can vary depending on the slope aspect (Li et al., 2021; Sharma et al., 2010). On different slope aspects, TN, TP, and the comprehensive soil nutrient index of surface soil were highest on the eastern and southern slopes, while the soil organic carbon content was highest on the northern slope. Plants need to absorb a large amount of fast-acting nitrogen and phosphorus during vegetative growth, and the nutrients required for plant growth are converted from organic matter in the soil. The lowest SOC, TN, TP, and the comprehensive soil nutrient index on the western slope are due to the fact that it was located in the wind–water erosion zone of the northern agro-pastoral ecotone, and the topsoil has been lost due to long-term wind erosion. The effect of different slope aspect conditions on soil pH was limited. This is because plant root systems and sediments were not abundant in the case of vegetation restoration of just 12a (Bai et al., 2020). The organic acid content was low when combined with organic matter during decomposition and vegetation restoration; therefore, it was insufficient to lower the pH of the surface soil (Seddaiu et al., 2013). Because controlling wind speed is the key to soil nutrient enhancement, future restoration projects that take place in dry alpine areas (i.e., the western and northern slopes) should prioritize the use of thickened non-woven fabric of at least 50 g $m^2$ for better insulation and to block wind, which is conducive to seed germination and seedling growth.

**4.3. Effects of slope aspect on soil erodibility**

Soil erodibility is commonly used to characterize the susceptibility of soils to water erosion and is influenced by vegetation and soil characteristics. Our results show that slope aspect has a significant effect on single soil erodibility indexes as well as comprehensive soil erodibility index. In general, soil erodibility decreases from the western slope to the eastern slope (Table 2), a pattern that may be related to the geographical location, altitude, temperature, and semi-arid climate of the region. Due to the location of our study site in the northern agro-pastoral ecotone of China, the western and northern slopes are susceptible to year-round gales from the northwestern




interior and Siberia, resulting in varying environmental conditions on the different
slope aspects. However, the soil water content of the northern slope (shaded slope) is
higher than that of the western slope, which may be more favorable for vegetation
restoration on the northern slope (Liu et al., 2020); the western slope may be more
vulnerable to erosion. Wind speed and soil moisture are key factors controlling the
process of vegetation restoration (Hupet and Vanclooster, 2002; Meng et al., 2018),
and these factors further influence soil erodibility (Sun et al., 2016). Therefore, future
studies should investigate methods to enhance vegetation restoration while utilizing
soil water resources available on the different slope aspects and reducing soil
erodibility.
**4.4. Relationship between soil nutrients and soil erodibility**
The comprehensive soil nutrient index was significantly positively correlated
with saturated hydraulic conductivity, mean weight diameter, and soil structure
stability index (Fig. 7), while the comprehensive soil nutrient index was highly
significantly negatively correlated with the comprehensive soil erodibility index, with
an explanation of 88% (Table S1). Many previous studies have reported similar
results (Dong et al., 2022a; Zhu et al., 2018). In this study, higher saturated hydraulic
conductivity, mean weight diameter, and soil structure stability index and lower soil
disintegration rate, K, and SOC cementing agent index indicate better soil structure
and lower soil erodibility. These characteristics can significantly reduce runoff and
sediment loss, which can result in soil nutrient accumulation (Pan and Shangguan,
2006; Sun et al., 2015; Zheng et al., 2021). Therefore, revegetation increases soil
nutrients and reduces soil erodibility, which further change vegetation and soil
characteristics. In addition, these factors could reduce soil nutrient loss and further
promote soil nutrient accumulation by reducing soil erodibility.
The comprehensive soil erodibility index was highly significantly negatively
correlated with SOC, TN, and TP (Fig. 7). Previous studies have shown that soil
organic matter and SOC are closely related to soil erodibility (Wang et al., 2019b).
SOC acts as a cement for soil aggregation, which improves soil structural stability
through the formation of aggregates, thus reducing soil erodibility. Soil nitrogen
indirectly affects soil erodibility by promoting plant growth and development,
increasing the accumulation of SOC in plants. In addition, nitrogen enrichment
increased soil macroparticles and mean weight diameter, which directly affected soil
erodibility. Similar to nitrogen, phosphorus is one of the essential elements for plant



growth and development, and the phosphorus content of soil determines the development of soil microorganisms and root systems, which will further influence the input of soil organic carbon and the formation of soil aggregates.

**4.5. Key factors impacting soil and vegetation related to slope aspect**

The interaction between soil and vegetation in the study area is complex. Because in the early stages of vegetation recovery, soil factors are unstable and vegetation is in the adaptation stage (Peng et al., 2009). The results derived from the random forest method showed that mean weight diameter, TP, saturated hydraulic conductivity, and soil disintegration rate were the main factors influencing the surface soil indicators. The main adhesion agents for the formation of aggregates included clay content, SOC and cementation. The mean weight diameter was significantly and positively correlated with soil organic carbon and clay content. The magnitude of mean weight diameter affects soil structural stability and root establishment, which varies due to environmental factors on different slope aspects. Soil phosphorus is an important element necessary for plant growth and development, and rapid growth requires more soil phosphorus, so there were some differences between different land use types on different slope aspects. The difference in TP between slope aspect affected the amount of inorganic phosphorus available for uptake by plants, and the lower phosphorus content limited plant growth. Saturated hydraulic conductivity reflects the permeability of soil and is an important indicator of soil erodibility. Differences in aboveground and belowground biomass of different slope aspects lead to different soil root traits, which affect the magnitude of saturated hydraulic conductivity. The soil disintegration rate was significantly negatively correlated with soil organic carbon, clay content, and mean weight diameter, and differences in soil microbial, nutrient, and root characteristics between slope aspects resulted in significant variations in the soil disintegration rate. By analyzing the main factors influencing surface soil quality in different slope aspects, timely application of phosphorus fertilizer in vegetation restoration projects could help accelerate the process of afforestation.

**4.6. Optimal land use type and plant species based on slope aspect**

Our study has shown that vegetation restoration can be an effective measure to improve soil nutrients and reduce soil erodibility. Moreover, the restored land use types and plant species to improve soil quality differed significantly depending on the slope aspect. Therefore, according to the differences in water, heat, wind, and sand on





different slope aspects in the northern agro-pastoral ecotone of China, the selection of
land use and its corresponding vegetation types should be carefully considered when
planning restoration projects to improve soil quality. The comprehensive soil nutrient,
erodibility, and quality indexes were established with a comprehensive investigation
of various soil nutrient and erodibility indexes. The optimal types of vegetation
restoration for different slope aspects was clarified. Our findings both agree with and
differ from previous studies (Colgan et al., 2010; Dong et al., 2022a; Wang et al.,
2021a). Studies that found contrasting results are likely due to the environmental
conditions (e.g. climate, rainfall, topographic conditions, seed bank, soil texture) of
the different slopes aspects. It is noteworthy that herbaceous vegetation on the
western slope is prone to severe shallow nutrient loss and soil erosion because of
strong wind conditions and sandy soil (Guo et al., 2020). Therefore, the use of
herbaceous vegetation should be carefully considered as the primary restoration
vegetation species. Fortunately, our proposal (*Caragana korshinskii* and *Lespedeza*
*bicolor*) satisfied this requirement. In addition, wind also contributes to soil erosion in
this region; however, limited research has been conducted on wind erosion and
combined erosion by wind and water. Future studies should be conducted on
combined erosion by wind and water study to better characterize soil erosion.
**5. Conclusions**

We found that some understory vegetation characteristics and soil properties

varied significantly with slope aspect. Soil nutrients and erodibility reflected by soil
organic carbon, total nitrogen, total phosphorus, saturated hydraulic conductivity, soil
disintegration rate, mean weight diameter, soil structure stability index, soil erodibility
factor, and soil organic carbon cementing agent index, respectively, were also
influenced by slope aspect and land use. Furthermore, comprehensive soil nutrient,
erodibility, and quality indexes also varied significantly with slope aspect, land use,
and predominant plant species. Slope aspect strongly modified the relationship
between comprehensive soil nutrient, erodibility, and quality indexes as well as
understory vegetation characteristics and soil properties. Our study found that
*Caragana korshinskii* and *Lespedeza bicolor* were the best taxa to include on any
slope aspect to improve soil nutrients and prevent soil erosion. This study provides
insight into the rational planning of vegetation restoration measures on all slope
aspects in the northern agro-pastoral ecotone in semi-arid areas.
**Date Availability**



Data will be made available on request.

**Author contributions.**

Yuxin Wu: Writing-original draft. Guodong Jia: Project administration, Funding acquisition, Writing-review and editing. Xinxiao Yu: Project administration, Funding acquisition, Writing-review and editing. Honghong Rao: Methodology and Formal analysis. Xiuwen Peng: Investigation. Yusong Wang: Investigation.

**Competing interests.**

The author declares that the publication of this scientific paper has no conflict of interest.

**Acknowledgements.**

We are grateful for the grants from the National Key Research and Development Program of China (2022YFF1302502-03) (China) and the National Natural Science Foundation of China (42230714).

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

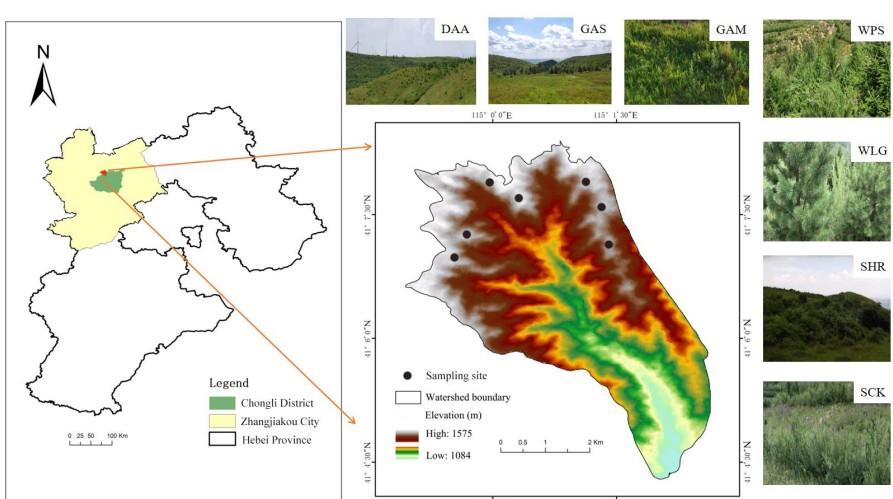


**Fig. 1.** Location map of the sampling points in the study area. The first letter: D, G, S and W represent degraded land, grassland, shrubland and woodland. The sampling sites from west to east were: DAA, degraded land; GAS, *Artemisia sacrorum*; GAM, *Astragalus melilotoides*; WPS, *Pinus sylvestris*; WLG, *Larix gmelinii*; SHR, *Hippophae rhamnoides*; SCK, *Caragana korshinskii*.

901

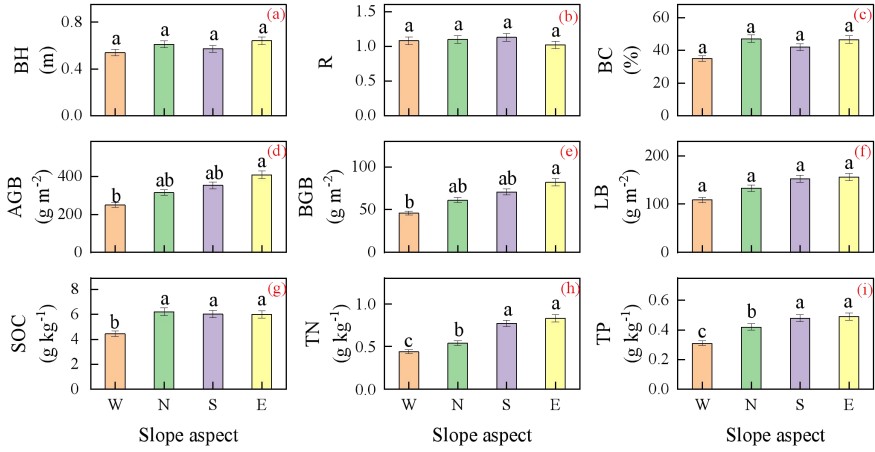

902

**Fig. 2.** Variation of understory vegetation characteristics and soil nutrients with slope aspects. BH, biomass height; R, richness; BC, biomass coverage; AGB, aboveground biomass; BGB, belowground biomass; LB, litter biomass; SOC, soil organic carbon; TN, total nitrogen; TP, total phosphorus; W, west; N, north; S, south; E, east. Different letters indicate significant differences among different seasons at $P<0.05$ level.

908



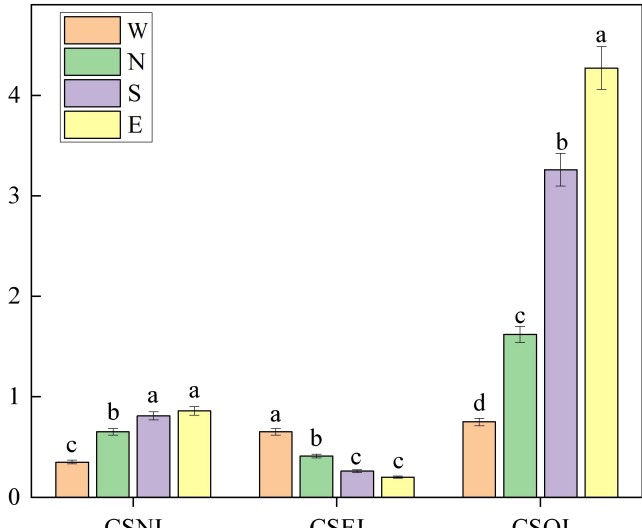

**Fig.3.** Variation of comprehensive soil nutrient, erodibility and quality index with slope aspects. CSNI, comprehensive soil nutrient index; CSEI, comprehensive soil erodibility index; CSQI, comprehensive soil quality index. Different letters indicate significant differences among different slope aspects at $P<0.05$ level.

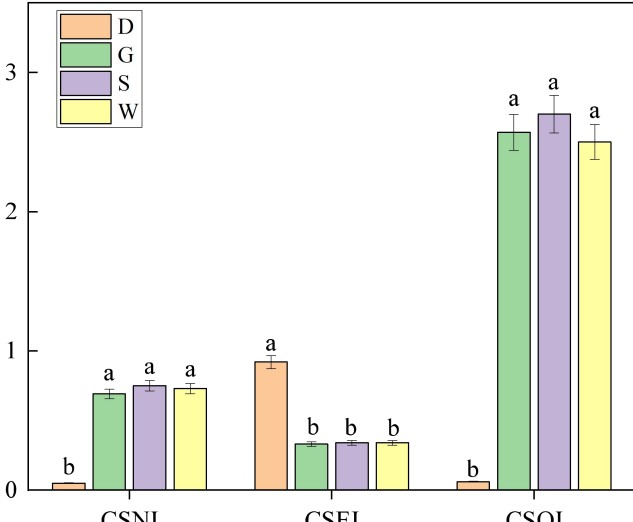

**Fig. 4.** Variation of comprehensive soil nutrient, erodibility and quality index with land use. Different letters indicate significant differences among different land use types at $P<0.05$ level.

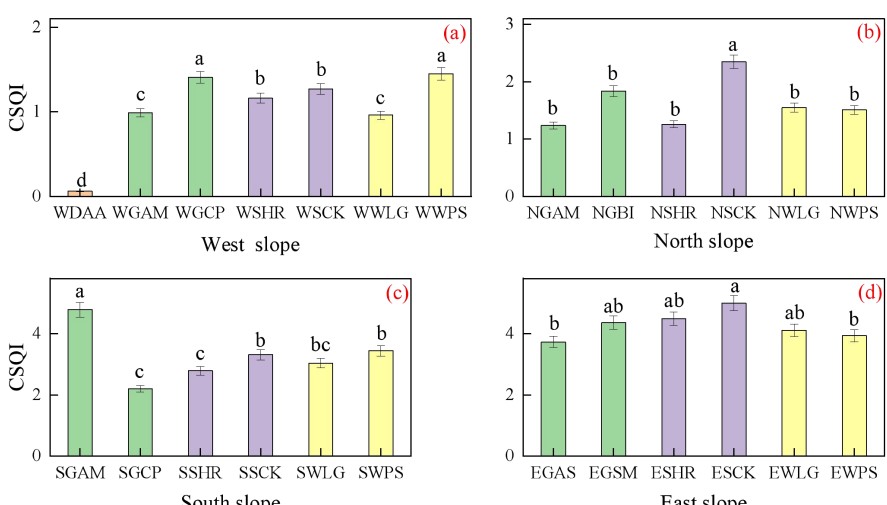

**Fig. 5.** Variation in comprehensive soil quality index with vegetation types along slope aspects. WDAA, *Artemisia annua*; WGAM, NGAM and SGAM, *Astragalus melilotoides*; NGBI, *Bothriochloa ischaemum*; EGSM, *Artemisia sacrorum, Astragalus melilotoides*; WGCP, NGCP and SGCP, *Capillipedium parviflorum*; WSHR, NSHR, SSHR and ESHR, *Hippophae rhamnoides*; WSCK, NSCK, SSCK and ESCK, *Caragana korshinskii*; WWLG, NSWG, SSWG and ESWG, *Larix gmelinii*; WWPS, NWPS, SWPS and EWPS, *Pinus sylvestris*. Different letters indicate significant differences among different seasons at *P*<0.05 level.

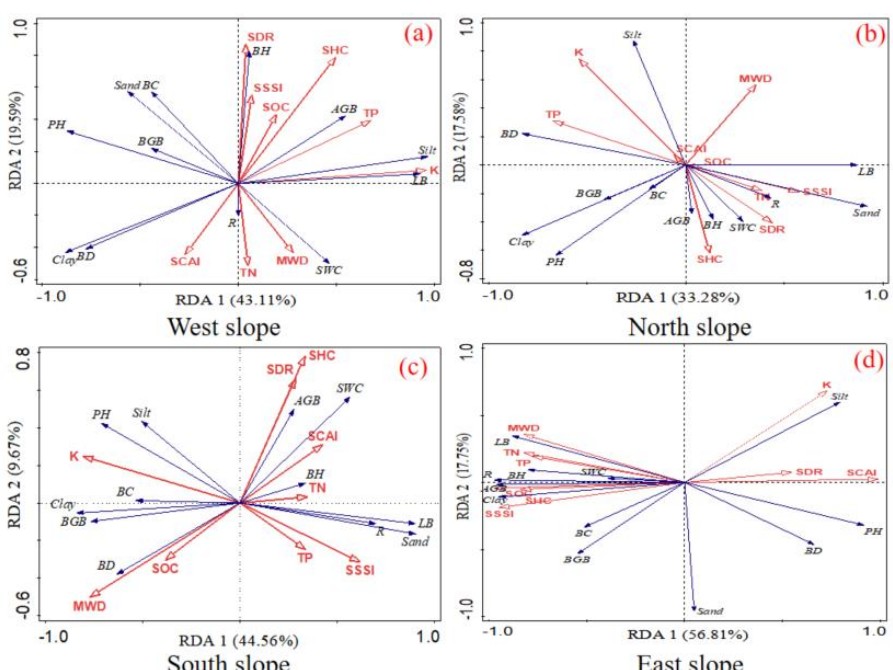

**Fig. 6.** Results of redundancy analysis (RDA) among soil quality parameters and characteristics of vegetation and soil on four slope aspects. BH: biome height; R: richness; BC: biome coverage; AGB: aboveground biomass; BGB: belowground biomass; LB: litter biomass; Sand: sand content; Silt: silt content; Clay: clay content; SWC: soil water content; SBD: soil bulk density; SOC: soil organic carbon; TN: total nitrogen; TP: total phosphorus; SHC, saturated hydraulic conductivity; SDR, soil disintegration rate; MWD, mean weight diameter; K, soil erodibility factor; SSSI, soil structure stability index; SCAI, SOC cementing agent index.



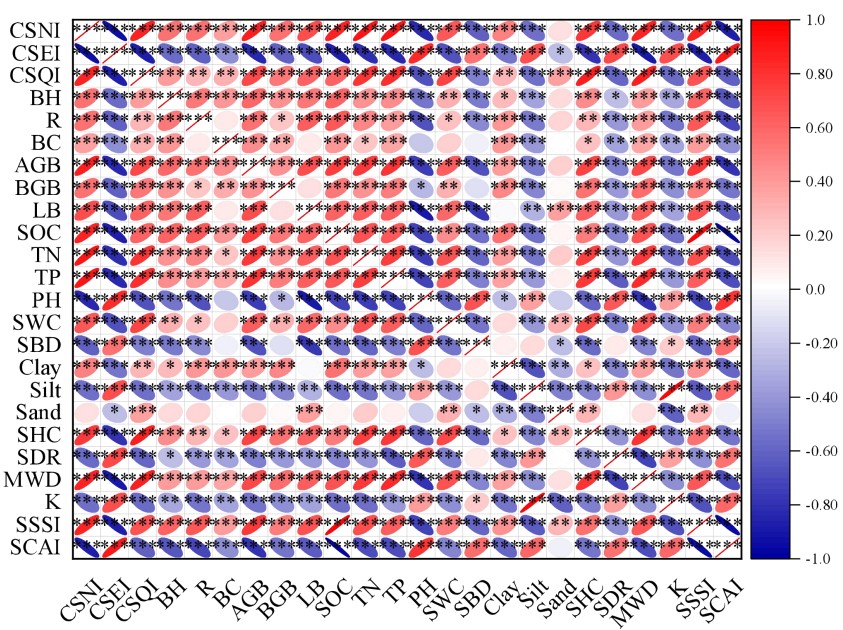

938
939

**Fig. 7.** Correlation analysis of CSNI, CSEI and CSQI with vegetation and soil characteristics. Red indicates a positive correlation, blue indicates a negative correlation, and the color depth indicates Pearson coefficients *$p < 0.05$, **$p < 0.01$ and ***$p < 0.001$, n = 84. CSNI, comprehensive soil nutrient index; CSEI, comprehensive soil erodibility index; CSQI, comprehensive soil quality index.

**Table 1**

The two-way ANOVA result for soil nutrient and erodibility. SOC: soil organic carbon; TN: total nitrogen; TP: total phosphorus; CSNI: comprehensive soil nutrient index; SHC: saturated hydraulic conductivity; SDR: soil disintegration rate; MWD: mean weight diameter; K: soil erodibility factor; SSSI: soil structure stability index; SCAI: SOC cementing agent index; CSEI: comprehensive soil erodibility index; CSQI: comprehensive soil quality index.

| soil variables | Land use type | | Slope aspect | | Land use ×Slope aspect | |
|---|---|---|---|---|---|---|
| Soil nutrient | F | P | F | P | F | P |
| SOC | 1200.37 | 0.000 | 50.985 | 0.000 | 5.818 | 0.000 |



| | | | | | | |
|---|---|---|---|---|---|---|
| TN | 520.016 | 0.000 | 79.681 | 0.000 | 24.354 | 0.000 |
| TP | 382.353 | 0.000 | 6.718 | 0.000 | 6.764 | 0.000 |
| CSNI | 832.059 | 0.000 | 46.447 | 0.000 | 6.851 | 0.000 |
| Soil erodibility | | | | | | |
| SHC | 824.538 | 0.000 | 54.173 | 0.000 | 52.672 | 0.000 |
| SDR | 799.513 | 0.000 | 6.632 | 0.001 | 3.956 | 0.000 |
| MWD | 1667.15 | 0.000 | 180.654 | 0.000 | 10.673 | 0.001 |
| K | 859.009 | 0.000 | 14.423 | 0.000 | 23.822 | 0.000 |
| SSSI | 517.098 | 0.000 | 41.05 | 0.000 | 26.717 | 0.000 |
| SCAI | 693.653 | 0.000 | 15.553 | 0.000 | 6.623 | 0.000 |
| CSEI | 1120.468 | 0.000 | 38.983 | 0.000 | 6.369 | 0.000 |
| Soil quality | | | | | | |
| CSQI | 642.05 | 0.000 | 103.399 | 0.000 | 35.679 | 0.000 |


**Table 2**
Soil erodibility indicators of different land use types at different slope aspect (mean
$\pm$ SD). SHC, saturated hydraulic conductivity; SDR, soil disintegration rate; MWD,
mean weight diameter; K, soil erodibility factor; SSSI, soil structure stability index;
SCAI, SOC cementing agent index. Different capital letters indicate significant
differences between slope aspects ($p<0.05$), different lowercase letters indicate
significant differences between the land use types ($p<0.05$).

| Slope aspect | Land use | SHC mm min$^{-1}$ | SDR g min$^{-1}$ | MWD mm | K t·hm$^2$·h·hm$^{-2}$· MJ$^{-1}$·mm$^{-1}$ | SSSI g kg$^{-1}$ | SCAI mm kg$^{-1}$ g$^{-1}$ |
|---|---|---|---|---|---|---|---|
| W | Degraded land | 0.13±0.02cC | 1.64±0.19aA | 0.79±0.02dD | 0.33±0.01aA | 0.25±0.01dB | 20.23±0.81aA |
| | grassland | 0.28±0.04bC | 0.29±0.04cA | 1.83±0.06bD | 0.26±0.01dA | 0.51±0.06bB | 9.09±0.97bA |
| | shrubland | 0.32±0.07bC | 0.82±0.53bA | 2.38±0.32aD | 0.32±0.01bA | 0.46±0.04cB | 9.03±0.80bA |
| | Woodland | 0.53±0.06aC | 1.58±0.07aA | 1.46±0.15cD | 0.27±0.01cA | 0.61±0.05aB | 7.53±0.70cA |
| N | grassland | 0.28±0.03bB | 0.26±0.02cB | 2.32±0.47bC | 0.31±0.01aAB | 0.50±0.06aA | 8.30±0.94aB |



| | | | | | | | |
|---|---|---|---|---|---|---|---|
| | shrubland | 0.31±0.04bB | 0.73±0.44bB | 2.84±0.12aC | 0.29±0.04aAB | 0.58±0.08aA | 8.14±0.95aB |
| | Woodland | 0.60±0.07aB | 1.26±0.17aB | 1.76±0.29cC | 0.29±0.01aAB | 0.57±0.03aA | 7.90±0.39aB |
| | grassland | 0.93±0.11bA | 0.24±0.01cBC | 3.28±0.04aB | 0.25±0.01cB | 0.51±0.10bA | 9.16±1.74aB |
| S | shrubland | 1.31±0.20aA | 0.40±0.11bBC | 3.32±0.06aB | 0.31±0.01aB | 0.53±0.03bA | 8.27±0.40abB |
| | Woodland | 1.45±0.14aA | 1.17±0.06aBC | 3.25±0.07aB | 0.28±0.01bB | 0.67±0.10aA | 6.94±1.00bB |
| | grassland | 1.55±0.18aA | 0.24±0.01cC | 4.06±0.14aA | 0.29±0.01aB | 0.59±0.02bA | 7.28±0.29bB |
| E | shrubland | 1.71±0.06aA | 0.31±0.07bC | 3.46±0.09bA | 0.26±0.02bB | 0.61±0.05bA | 8.18±0.89aB |
| | Woodland | 1.73±0.12aA | 0.38±0.03aC | 3.42±0.10bA | 0.28±0.01bB | 0.71±0.05aA | 6.41±0.44cB |
