# Peer review of "Response of soil nutrients and erodibility to slope aspect in the northern agro-pastoral ecotone, China"

_EGUsphere, 2023_

## Author Comment (AC2)

**Table S1** Understory vegetation characteristics of different vegetation types at different slope aspect. Values are in the form of the mean ± standard error. Different capital letters indicate significant differences between slope aspects (p<0.05),different lowercase letters indicate significant differences between the four vegetation types (p<0.05).

| Slope aspect | Land use | Site code | Dominated plant communities | Dominant herbaceous plant | Biomass height (m) | Richness | Biomass coverage (%) | Aboveground biomass (g m-2) | Belowground biomass (g m-2) | Litter biomass (g m-2) |
|---|---|---|---|---|---|---|---|---|---|---|
| W | Degraded land | WDAA | Artemisia annua | Potentilla chinensis,Artemisia annua | 0.24±0.02aA | 0.71±0.04aA | 20.11±6.88bA | 55.94±3.52aB | 26.48±4.39bB | 54.49±4.42dA |
| | grassland | WGAM | Astragalus melilotoides | Astragalus melilotoides | 0.45±0.08aA | 1.15±0.02aA | 49.42±20.53aA | 270.88±56.11aB | 39.19±22.50aB | 96.92±24.63cA |
| | | WGCP | Capillipedium parviflorum | Capillipedium parviflorum | 0.55±0.11aA | 1.12±0.09aA | 47.61±24.66aA | 253.4±30.09aB | 73.92±17.54aB | 96.45±17.50cA |
| | shrubland | WSHR | Hippophae rhamnoides | Artemisia sacrorum,Capillipedium parviflorum | 0.54±0.17aA | 1.14±0.06aA | 28.67±4.73bA | 236.15±36.57aB | 26.61±9.87bB | 141.5±10.99aA |
| | | WSCK | Caragana korshinskii | Artemisia sacrorum,Capillipedium parviflorum | 0.67±0.39aA | 1.13±0.02aA | 30.94±17.13bA | 312.83±8.12aB | 29.65±1.09bB | 135.2±10.50aA |
| | Woodland | WWLG | Larix gmelinii | Astragalus melilotoides, Artemisia sacrorum | 0.71±0.05bA | 1.17±0.07bA | 35.27±18.32bA | 268.23±30.21bB | 76.71±6.07aB | 122±7.99bA |
| | | WWPS | Pinus sylvestris | Astragalus melilotoides, Artemisia sacrorum | 0.63±0.14bA | 1.11±0.04bA | 32.67±13.05bA | 350.14±12.38bB | 49.17±13.71aB | 113.46±7.43bA |
| N | grassland | NGAM | Astragalus melilotoides | Astragalus melilotoides | 0.55±0.09aA | 1.02±0.17aA | 37.79±1.91aA | 292.97±62.32aAB | 62.75±20.67aAB | 95.08±31.90cA |
| | | NGBI | Bothriochloa ischaemum | Bothriochloa ischaemum (L.) Keng | 0.56±0.06aA | 1.01±0.17aA | 67.41±1.28aA | 282.81±70.70aAB | 79.76±12.14aAB | 89.55±13.09cA |
| | shrubland | NSHR | Hippophae rhamnoides | Astragalus melilotoides,Potentilla chinensis | 0.72±0.06aA | 1.2±0.11aA | 56.78±20.08aA | 305.83±19.11aAB | 46.69±20.66aAB | 175.16±12.81aA |

| | | | | | | | | | | |
|---|---|---|---|---|---|---|---|---|---|---|
| S | Woodland | NSCK | Caragana korshinskii | Artemisia sacrorum,Capillipedium parviflorum | 0.41±0.05aA | 1.09±0.04aA | 25.11±6.71aA | 300.94±33.44aAB | 43.08±8.75aAB | 167.37±12.24aA |
| | | NWLG | Larix gmelinii | Artemisia sacrorum,Capillipedium parviflorum | 0.77±0.06aA | 1.12±0.08aA | 31.79±28.25aA | 295.86±32.64aAB | 88.27±6.15aAB | 140.2±16.10bA |
| | | NWPS | Pinus sylvestris | Astragalus melilotoides | 0.63±0.05aA | 1.13±0.02aA | 63.67±14.98aA | 411.27±49.26aAB | 47.07±9.84aAB | 130.38±14.98bA |
| | grassland | SGAM | Astragalus melilotoides | Astragalus melilotoides | 0.51±0.08bA | 1.12±0.01aA | 43.46±13.38abA | 304.11±14.56bAB | 77.84±42.56aAB | 110.62±23.76cA |
| | | SGCP | Capillipedium parviflorum | Capillipedium parviflorum | 0.42±0.15bA | 1.18±0.09aA | 40.55±12.51abA | 276.32±63.54bAB | 76.38±49.01aAB | 108.54±6.02cA |
| | shrubland | SSHR | Hippophae rhamnoides | Artemisia sacrorum,Potentilla chinensis | 0.66±0.12bA | 1.19±0.11aA | 28.33±18.58bA | 397.55±19.17aAB | 56.71±6.77aAB | 207.31±14.62aA |
| | | SSCK | Caragana korshinskii | Capillipedium parviflorum,Lespedeza bicolor | 0.41±0.05bA | 1.1±0.03aA | 25.11±5.42bA | 361.4±11.68aAB | 47.7±7.11aAB | 198.08±13.97aA |
| | Woodland | SWLG | Larix gmelinii | Artemisia sacrorum | 0.75±0.01aA | 1.09±0.12aA | 54.67±21.36aA | 317.5±20.12aAB | 91.97±3.46aAB | 149.14±12.11bA |
| | | SWPS | Pinus sylvestris | Astragalus melilotoides,Artemisia sacrorum | 0.69±0.10aA | 1.1±0.11aA | 60±17.58aA | 459.27±38.92aAB | 73.73±7.92aAB | 138.7±11.27bA |
| E | grassland | EGAS | Astragalus melilotoides | Artemisia sacrorum | 0.54±0.12aA | 0.9±0.06aA | 55.87±14.29aA | 337.29±56.74bA | 109.63±18.71aA | 106.87±15.61cA |
| | | EGSM | Astragalus melilotoides | Artemisia sacrorum | 0.55±0.12aA | 0.96±0.11aA | 51.75±23.80aA | 350.39±37.68bA | 103.14±3.28aA | 114.63±2.93cA |
| | shrubland | ESHR | Hippophae rhamnoides | Artemisia sacrorum,Potentilla chinensis | 0.7±0.24aA | 1.27±0.08aA | 37.22±6.74aA | 428.69±34.74abA | 63.33±3.28cA | 214.75±32.17aA |
| | | ESCK | Caragana korshinskii | Capillipedium parviflorum,Lespedeza bicolor | 0.69±0.38aA | 1.05±0.01aA | 35.42±17.95aA | 414.61±34.58abA | 49.49±13.33cA | 205.19±30.74aA |

| | | | | | | | | | |
|---|---|---|---|---|---|---|---|---|---|
| Woodland | EWLG | Larix gmelinii | Artemisia sacrorum | 0.77±0.11aA | 1.03±0.25aA | 40.1±12.90aA | 364.47±53.42aA | 92.56±5.59bA | 152.01±9.17bA |
| | EWPS | Pinus sylvestris | Astragalus melilotoides | 0.6±0.15aA | 0.9±0.22aA | 59±13.89aA | 552.13±32.97aA | 73.8±13.84bA | 141.37±8.53bA |

**Table S2** Soil characteristics of different vegetation types at different slope aspect. Values are in the form of the mean ± standard error. SWC: soil water content; SBD: soil bulk density; SOC: soil organic carbon; TN: text Normalization; TP: total phosphorus. Different capital letters indicate significant differences between slope aspects ($p<0.05$), different lowercase letters indicate significant differences between the four vegetation types ($p<0.05$).

| Slope aspect | Land use | Site code | SWC (%) | BD (g cm-3) | Clay (%) | Silt (%) | Sand (%) | pH |
|---|---|---|---|---|---|---|---|---|
| W | Degraded land | WDAA | 0.08±0.03cC | 1.63±0.03aA | 10.25±0.23dD | 9.12±0.12bB | 80.63±0.32abAB | 8.5±0.11aA |
| | grassland | WGAM | 0.1±0.01bcC | 1.58±0.01aA | 13.63±0.09bD | 5.22±0.17cB | 81.15±0.14aAB | 7.6±0.19bA |
| | | WGCP | 0.1±0.03bcC | 1.66±0.03aA | 13.19±0.09bD | 5.52±0.17cB | 81.29±0.14aAB | 7.44±0.19bA |
| | shrubland | WSHR | 0.11±0.02abC | 1.25±0.02bA | 12.42±0.23cD | 8.36±0.21bB | 79.21±0.28bAB | 6.97±0.07cA |
| | | WSCK | 0.13±0.01abC | 1.23±0.03bA | 12.86±0.23cD | 8.06±0.21bB | 79.07±0.28bAB | 6.99±0.07cA |
| | Woodland | WWLG | 0.12±0.01aC | 1.26±0.02bA | 20.11±0.34aD | 18.77±0.43aB | 61.12±0.77cAB | 7.53±0.02bA |
| | | WWPS | 0.16±0.01aC | 1.24±0.01bA | 20.55±0.26aD | 18.47±0.17aB | 60.98±0.43cAB | 7.55±0.02bA |
| N | grassland | NGAM | 0.1±0.03bBC | 1.56±0.01aA | 13.45±0.36bB | 7.49±0.33bC | 79.06±0.28bA | 7.48±0.05aB |
| | | NGBI | 0.1±0.01bBC | 1.63±0.13aA | 13.01±0.36bB | 7.79±0.33bC | 79.2±0.28bA | 7.33±0.05aB |
| | shrubland | NSHR | 0.12±0.04abBC | 1.22±0.13bA | 10.79±0.46cB | 7.62±1.88bC | 81.59±1.45cA | 6.83±0.10bB |
| | | NSCK | 0.12±0.02abBC | 1.2±0.07bA | 11.23±0.46cB | 7.32±1.88bC | 81.45±1.45cA | 6.85±0.10bB |
| | Woodland | NWLG | 0.13±0.01aBC | 1.25±0.07bA | 18.38±0.08aB | 28.19±0.28aC | 53.43±0.29aA | 7.48±0.05aB |
| | | NWPS | 0.16±0.01aBC | 1.22±0.02bA | 18.82±1.00aB | 27.89±0.28aC | 53.29±0.72aA | 7.5±0.05aB |
| S | grassland | SGAM | 0.1±0.01bAB | 1.53±0.02aB | 14.73±0.22bC | 4.45±0.39cB | 80.82±0.61aB | 7.35±0.04aC |

|  |  |  |  |  |  |  |  |
|---|---|---|---|---|---|---|---|
|  |  | SGCP | 0.11±0.02bAB | 1.6±0.08aB | 14.29±0.22bC | 4.75±0.39cB | 80.96±0.61aB | 7.2±0.03aC |
|  | shrubland | SSHR | 0.12±0.02aAB | 1.21±0.08bB | 11.56±0.09cC | 8.03±0.15bB | 80.42±0.24bB | 6.56±0.04bC |
|  |  | SSCK | 0.13±0.01aAB | 1.19±0.13bB | 12±0.09cC | 7.73±0.15bB | 80.28±0.24bB | 6.58±0.04bC |
|  | Woodland | SWLG | 0.13±0.03aAB | 1.24±0.13bB | 20.09±0.23aC | 29.93±0.29aB | 49.98±0.13cB | 7.29±0.05aC |
|  |  | SWPS | 0.16±0.01aAB | 1.22±0.07bB | 20.53±0.35aC | 29.63±0.29aB | 49.84±0.62cB | 7.3±0.05aC |
| E | grassland | EGAS | 0.11±0.01aA | 1.34±0.11aB | 13.44±0.13bA | 6.55±0.12bA | 80.01±0.01bC | 7.16±0.02bB |
|  |  | EGSM | 0.11±0.02aA | 1.41±0.12aB | 13±0.13bA | 6.85±0.12bA | 80.15±0.01bC | 7.01±0.01bB |
|  | shrubland | ESHR | 0.11±0.01abA | 1.14±0.09bB | 10.65±1.04cA | 6.15±1.22bA | 83.2±1.30aC | 6.34±0.27cB |
|  |  | ESCK | 0.14±0.02abA | 1.12±0.05bB | 11.09±1.04cA | 5.85±1.22bA | 83.06±1.30aC | 6.36±0.27cB |
|  | Woodland | EWLG | 0.13±0.01bA | 1.2±0.03bB | 18.94±0.20aA | 25.01±0.22aA | 56.05±0.02cC | 7.26±0.04aB |
|  |  | EWPS | 0.17±0.01bA | 1.17±0.03bB | 19.38±0.20aA | 24.71±0.22aA | 55.91±0.02cC | 7.28±0.04aB |

Profile figure 1-3

---

## Author Response (AR1)

Dear Editors and Reviewers:

Thank you for your letter and for the reviewers' comments concerning our manuscript entitled "Response of soil nutrients and erodibility to slope aspect in the northern agro-pastoral ecotone, China" (ID: EGUSPHERE-2023-1006). Those comments are all valuable and very helpful for revising and improving our paper, as well as the important guiding significance to our researches. We have studied comments carefully and have made correction which we hope meet with approval. Revised portion has been marked in red in the paper. The main corrections in the paper and the responds to the reviewer's comments are as flowing:

Responds to the reviewer's comments:

Reviewer #1: I do not have any remarks. The manuscript is not so innovative but the results are applicable in practice. Congratulations to the authors!

Response: Thank you very much for your valuable comments on our paper.

Reviewer #2: your work is interesting. Some parts are complete. The introduction adequately describes the problem and the state of the art. However, several issues need to solve and a clarification.

Response: Thank you very much for your valuable comments. At the same time, we responded to all issues accordingly.

- You declare a "difficult to find" degraded land (20 m x 20 m sample site) in your study area (ROW 166). This sentence could be important because influence the experimental design. Probably, you must clarify better this point, "the vast majority of the degraded land had been converted to artificial forest and grass vegetation" is not enough ( and without reference). Further, a more accurate definition of degraded land is necessary.

Response: We modified this part (ROW 166). What we originally wanted to say was that the area was well restored after vegetation reconstruction, that it was difficult to

find degraded land on each slope aspect at a particular slope gradients and positions, and that the western slope was partially unresolved due to the harsher natural environment. Therefore, we have chosen the degraded land on the western slope as a point of contrast. In addition, we have further defined degraded land as follows: Line 145-147"The degraded land (loss of soil material from wind and water erosion, degradation of physical, chemical and biological properties of soil) was previously degraded cropland."

- In your work, you analyze the role of the slope aspect in erodibility and soil nutrient availability. But, in the sample site selection, did you have taken into account the slope gradient? Could it be a driver of a statistical analysis? I think the sites must have comparable morphological and topographic conditions in order to perform statistical analysis.

Response: We agree with you very much. We took this detail into account in the field survey. For example:"in the sample site selection, the slope gradients and positions were similar for all selected sample sites." (ROW 168). Furthermore, We also think the sites must have comparable morphological and topographic conditions in order to perform statistical analysis.

- Please, provide further information about the soil (and geological too). Did you have used a soil map to define soil? Please, cite it. Provide a complete and correct WRB soil classification, a generic "chestnut soil" is very poor as a soil classification for the scale of your work.  Did you open some soil profiles? further information could be necessary   (depth, horizons, ...).

Response: The soil characteristics of different vegetation types at different slope aspect were shown in table S2, including soil water content; soil bulk density, soil texture (Clay, Silt and Sand), soil organic carbon, total Normalization, and total phosphorus. In addition, we dug and sampled each sample point, part of the diagram can be seen in the submission document (Fig. profile figure 1-3).

- The sites' position map is not so clear, the look like the sites are more or less at the same elevation but no further information emerges.

Response: The sites in the map correspond to vegetation diagrams from west to east (Line: 896-899 The sampling sites from west to east were: DAA, degraded land; GAS, *Artemisia sacrorum*; GAM, *Astragalus melilotoides*; WPS, *Pinus sylvestris*; WLG, *Larix gmelinii*; SHR, *Hippophae rhamnoides*; SCK, *Caragana korshinskii*). We initially wanted to mark each sample point on the map, but need to enlarge the map to each sample point. Too many sample points are prone to overlap, resulting in excessive, unsightly and messy diagrams. In addition, figure 1 was entirely created by the authors.

- You identify some species as the best candidates for restoration projects. I guess this choice regards soil loss. Which is the situation in this area concerning further natural hazards, such as wildfires and landslides, strictly connected to environmental conditions? Are reasonable to adopt a multi-risk approach in your study area?

Response: The area has corresponding rehabilitation measures, including grains, silt dams and slope protection works, can effectively prevent natural disasters. Future work will focus on land degradation associated with soil erosion from water and storms in the region. In the next step, we will conduct a comparative study of vegetation restoration, engineering restoration, and combined restoration measures. We believe that a multi-risk approach may be feasible.

**Profile figure 1-3**

[Figure]

[Figure]

Editor's comments.

Response to the editorial support team: We have modified the information of one author and added one author who contributed to this paper (line 4 and11). In addition, we have made an overall top-down revision of the paper, as detailed below:

Line 20 and 43: "impacting" to "impacts" and "influencing" to "influences"

Line 33: Understory vegetation and soil characteristics  explained 50.86–74.56% of the total variance in soil nutrients and erodibility of slope aspect.

Line 59-63: We found the logic of the following two sentences reversed and switched their order. "Most studies have only focused on one aspect; thus, they lack comprehensive consideration and evaluation of the impact of land use changes caused by vegetation restoration on soil nutrients and erodibility." and "However, it is not

clear which plants selected for restoration are the most effective in enhancing soil nutrients and reducing soil erodibility."

Line 110-114: Based on the abovementioned scientific gaps, we hypothesize that both slope aspect and land use types can significantly alter soil structure and properties to influence soil nutrients and erodibility under vegetation restoration. We further hypothesize that the western slope may have the lowest comprehensive soil quality index compared to other slope aspects.

Line 125: "It includes 28 sample sites (20 m × 20 m) of a degraded land, two grasslands..."

Line 164: "Following the methods described by Yimer (2006)..."

Line 246: " SPSS Ver. 20 software..."

Line 367-518: Consider that the discussion section is overloaded. We deleted some of the content and tightly centered on our hypothesis and answered the questions raised by the hypothesis through discussion. Details can be seen in word 20230908-in red.

Line 519: We have added at the end of the "Future work will focus on land degradation associated with soil erosion from water and storms in the region."

With kind regards,
Sinerely,
Guodong Jia

---

## Referee Report (RR1)

**Response of soil nutrients and erodibilty to slope aspect in the nother agro-pastoral ecotone, China**

**Review:**

**Abstract:**

Line 18: it is better to say how soil erosion is considered **one of** a major environmental and social problem… leads to the **changes** in **soil physico-chemical characteristic**, impacting plant growth. (Changes in soil characteristics do not implicate degradation).

Line 23: what is typical watershed?

**Introduction:**

Lines 42 and 43: Please, stick to the comments above.

Line 68: There is currently no soil erosion model that can accurately predict soil erosion rates…. I do not agree. What about radionuclides, which are the mostly recommended technique for estimation soil erosion losses in absolute amounts.

Line 82: which topographic factors? Yes, in the next sentence you mention slope (gradient and position of slope), but you need to precise which else topographic factors in brackets, and emphasize the slope as the most important.

Line 97:"……soil erosion in this zone is considered very serious". How many? Do you have approximate information in t ha-1 y-1?

Line 113: Please, explain why do you think that about hypothesis number 3.

Line 133: Please, put a reference in brackets about rock formation, any official map or reference as you did for the soil types before.

**Soil sampling and analysis:**

Generally, you need to mention the name of laboratory where the analysis were conducted.

**Key factors impacting soil and vegetation related to slope aspect:**

Please, throw out the first two sentences and focus on analyzing the results

Please, avoid (in whole manuscript) terminology: "Our study"…, better use "The results of this research" .... in passive construction.